# The application of allostasis and allostatic load in animal species: A scoping review

Kathryn E. Seeley[ID][1]*, Kathryn L. Proudfoot[2], Ashley N. Edes[ID][3]

**1** Department of Preventive Medicine, The Ohio State University College of Veterinary Medicine, Columbus, Ohio, United States of America, **2** Health Management, Atlantic Veterinary College, University of Prince Edward Island, Charlottetown, Prince Edward Island, Canada, **3** Department of Reproductive and Behavioral Sciences, Saint Louis Zoo, St. Louis, Missouri, United States of America

* seeley.katie@gmail.com

**Data Availability Statement:** All relevant data are within the paper and its Supporting Information files.

## Abstract

Principles of allostasis and allostatic load have been widely applied in human research to assess the impacts of chronic stress on physiological dysregulation. Over the last few decades, researchers have also applied these concepts to non-human animals. However, there is a lack of uniformity in how the concept of allostasis is described and assessed in animals. The objectives of this review were to: 1) describe the extent to which the concepts of allostasis and allostatic load are applied theoretically to animals, with a focus on which taxa and species are represented; 2) identify when direct assessments of allostasis or allostatic load are made, which species and contexts are represented, what biomarkers are used, and if an allostatic load index was constructed; and 3) detect gaps in the literature and identify areas for future research. A search was conducted using CABI, PubMed, Agricola, and BIOSIS databases, in addition to a complementary hand-search of 14 peer-reviewed journals. Search results were screened, and articles that included non-human animals, as well as the terms "allostasis" or "allostatic" in the full text, were included. A total of 572 articles met the inclusion criteria (108 reviews and 464 peer-reviewed original research). Species were represented across all taxa. A subset of 63 publications made direct assessments of allostatic load. Glucocorticoids were the most commonly used biomarker, and were the only biomarker measured in 25 publications. Only six of 63 publications (9.5%) constructed an allostatic load index, which is the preferred methodology in human research. Although concepts of allostasis and allostatic load are being applied broadly across animal species, most publications use single biomarkers that are more likely indicative of short-term rather than chronic stress. Researchers are encouraged to adopt methodologies used in human research, including the construction of species-specific allostatic load indexes.

## Introduction

The link between stress and health is well documented across several taxa [1–3]. Acute, short-term stress is adaptive and essential for survival [3]; however, chronic or prolonged stress can have significant impacts on morbidity and mortality [4–6]. Most of the research evaluating the

**Funding:** The author(s) received no specific funding for this work.

**Competing interests:** The authors have declared that no competing interests exist.

link between chronic stress and negative health outcomes has been done in humans [7–9] or laboratory animal models [10–12]. However, there is a growing body of work that evaluates the impact of chronic stress in non-human animals, both under managed care and in their native ranges [13–15].

Due to the important role of stress in human and animal health, many methods of measuring stress have been developed. These measurements include individual biomarkers such as leukocyte numbers and composition [16], leukocyte function [17], heart rate variability [18, 19], and glucocorticoids [20–22]. However, given that stressors cause a complex physiological and behavioral response in animals, there is no single measure that can fully quantify the stress response or its long-term effects; instead, it has been suggested that multiple biomarkers be used when making assessments about chronic stress [23–25].

In the last several decades, a concept known as "allostasis" has emerged as a framework for the complicated physiologic processes involved in the stress response [26, 27]. Allostasis is the idea of "stability through change" in which an organism makes physiological and behavioral adjustments in response to predictable and unpredictable stressors [26–28]. Allostasis is a complementary concept to homeostasis, which refers to the maintenance of certain physiological parameters within very narrow ranges [29]. Unlike homeostasis, allostatic parameters are not maintained within narrow ranges but instead fluctuate according to demand, such as an increase in heart rate and blood pressure during physical activity. Together, allostasis and homeostasis provide a holistic model for an organism's response to stressors. Allostasis is maintained using the integrated responses of physiological axes such as the hypothalamic-pituitary-adrenal (HPA) and sympathetic-adrenal-medullary (SAM) axes, allowing for adaptation to both internal and external stressors [26].

Allostatic load (AL) is the cumulative cost incurred by somatic systems due to repeated or chronically activated allostasis [26, 30, 31]. All organisms experience daily and seasonal stressors with which they need to cope [26, 28, 31]. For instance, a prey animal will alter its behavior when faced with a predator, allowing it to evade predation and removing the stressor. However, when repeated or ongoing stressors overload an animal, there can be "wear and tear" across multiple somatic systems, which predisposes individuals to physiologic dysregulation and subsequent poor health [32, 33].

Allostatic load itself cannot be directly measured, instead it can only be inferred based on quantifiable biomarkers that change (either increase or decrease) from baseline levels due to the systemic dysregulation that occurs with chronically or repeatedly activated allostasis [23]. In 1997, Seeman et al. developed a model that estimated AL through an index (hereafter referred to as an "allostatic load index," or ALI) that combined ten biomarkers to reflect the function of the neuroendocrine, cardiovascular, and metabolic systems. These initial biomarkers included norepinephrine, epinephrine, cortisol, dehydroepiandrosterone sulfate (DHEA-S), systolic blood pressure, diastolic blood pressure, high-density lipoprotein (HDL), total cholesterol-HDL ratio, waist-hip ratio, and glycosylated hemoglobin (HbA1c) [31]. This composite of biomarkers included both primary mediators of the stress response as well as the secondary mediators of allostasis. Since the original publication, over 50 different biomarkers have been used in various ALIs in the human literature [5, 23, 34].

ALIs are the primary means of evaluating AL in human populations, particularly in the fields of human health, anthropology, and sociology [34, 35]. There are hundreds of publications reporting associations between AL and chronic stressors in humans [4, 5, 36]. For example, it has been shown that people who disclose their sexual orientation have lower AL than those that do not [37], that caregiving and AL are predictive of future illness or disability [38], and that AL is impacted by ethnicity, gender, and educational attainment [39].

Several of the biomarkers incorporated into ALIs in human populations are well conserved across taxa. For instance, cortisol, glucose, DHEA-S, and interleukins are frequently incorporated into ALIs in humans [5] and are also measured in animal populations [40–42]. Despite the overlap in the individual biomarkers that are being measured in animals and humans, the application of ALIs to non-human animals has been limited, and individual biomarkers are sometimes used as proxies instead. In the last five years, our group has called for the application of a more rigorous AL methodology in animal populations [23, 43]. To better understand how the terms allostasis and AL have been used to date and where there are gaps in methodology, we conducted a scoping review of how the concepts of allostasis and AL are being applied in non-human animals.

## Materials and methods

### Review protocol and expertise

This scoping review was conducted using the Arksey and O'Malley framework [44]. The PRISMA-ScR checklist was utilized to ensure completeness (S1 Checklist). The protocol was created in advance of the literature review based on the input and expertise of the co-authors, which include veterinary medicine (KS), biological anthropology and primatology (AE), and animal welfare (KP). The repository of relevant articles and the resulting datasets are available upon request.

### Review question and scope

The overall goal of this review was to describe how principles of allostasis are being used in research with non-human animals. We had three primary objectives: 1) describe the extent to which the concepts of allostasis and AL are applied theoretically to animal populations, with a focus on which taxa and species are represented; 2) identify when direct assessments of allostasis or AL are made, which species and contexts are represented, what biomarkers are used, and if an ALI was constructed; and 3) detect gaps in the literature and identify areas for future research.

### Search strategy

A comprehensive search was conducted in CABI, PubMed®, Agricola, and BIOSIS™ databases on June 5, 2021, using the following algorithm: (allostasis OR allostatic AND (animal* OR wildlife* OR mammal* OR primate* OR avian* OR bird* OR reptile* OR snake* OR lizard* OR turtle* OR tortoise* OR amphibian* OR frog* OR fish*) NOT (human* OR hominidae OR "homo sapien")). There were no constraints on the publication date for this search.

Databases do not always search the full text, so a complementary hand-search of 14 peer-reviewed journals was done. To select which journals were hand-searched, the results of the initial database search were categorized by publication and placed in rank order. Initially, the top 10 journals were going to be hand-searched. However, the 10th ranked journal had the same number of publications as the journals ranked 11 to 14, so all 14 were included in the review. The journals that were hand-searched included: General and Comparative Endocrinology; Hormones and Behavior; Physiology & Behavior; Comparative Biochemistry and Physiology Part A: Molecular and Integrative Physiology; Functional Ecology; PLOS One; Aquaculture; Integrative and Comparative Biology; Scientific Reports; Animals; Conservation Physiology; Journal of Experimental Biology; Oecologia; and Physiological and Biochemical Zoology. The website for each journal was used to conduct a full text search for the terms "allostasis" or "allostatic" and retrieve all relevant publications.

### Relevance screening and inclusion criteria

The inclusion criteria for the scoping review were intentionally broad to allow for the inclusion of any relevant articles using non-human animal species. All search results were exported into Excel (Excel 2016, Microsoft Corporation, Redmond, WA). Duplicates were highlighted and removed using conditional formatting. An initial screening of the title and abstract was conducted by one author (KS). Articles written about non-human animals were included for full review, unless it was evident in the title or abstract that the animal was being used as a model for human disease. Only peer-reviewed research or review articles were included; editorials, commentaries, letters, conference proceedings, theses, and invited papers were excluded. Papers that were not available in English or could not be obtained via the Ohio State University Library system were also excluded.

After screening the title and abstract, all articles that were initially included were exported to a free citation manager (Zotero) and underwent full text review by one author (KS). Any article that pertained directly to a non-human animal population and included the terms "allostasis" or "allostatic" in the full text was included in the scoping review. Similar to the initial screening, any studies in which the animals were used as direct proxies for humans were excluded; however, laboratory animal studies were included if the research was aimed at learning about the welfare and response of the animal, and they were not used as models for humans. Articles where the search terms were listed in the literature cited or acknowledgments sections and not in other sections of the text were excluded. If the search terms were not found at all in the full text review, the publications were excluded.

To fulfill the second objective of this review, any publication that made direct assessments of AL by measuring biomarkers were coded by one author (KS). For the purpose of this review, only biomarkers that could be evaluated from ante-mortem samples (e.g., saliva, hair, feces, blood) or could be directly measured in living animals (e.g., heart rate, blood pressure) were evaluated.

### Scoping review management, data charting, and analysis

All articles that met the inclusion criteria had the following data recorded: 1) taxa and species; 2) where in the article the search terms were mentioned (e.g., introduction, results); 3) whether the article was a review (including meta-analyses) or primary research; 4) which biomarkers were measured; 5) whether AL was inferred based on the biomarkers measured; and 6) if an ALI was constructed to evaluate AL.

## Results

### Descriptive statistics

Following the literature search and removal of duplicates, 2,460 publications were identified for title and abstract review, out of which 1,212 articles were identified for full text review (Fig 1). Reasons for exclusion after full text review included animals being used as models for human disease (n = 63), conceptual articles that did not specifically apply to animal populations (n = 14), the terms "allostasis" or "allostatic" being found in the bibliography only (n = 553), or the terms not being present in the full text (n = 10). A total of 572 articles met all inclusion criteria and were included in this scoping review (S1 Dataset). Of the included articles, 108 were review articles (including meta-analyses) and 464 were peer-reviewed original research publications.

Articles were written between 2003 and 2021, with 84% (479/572) published within the last 10 years (Fig 2). Since the literature search was conducted in June of 2021, the bar in Fig 2

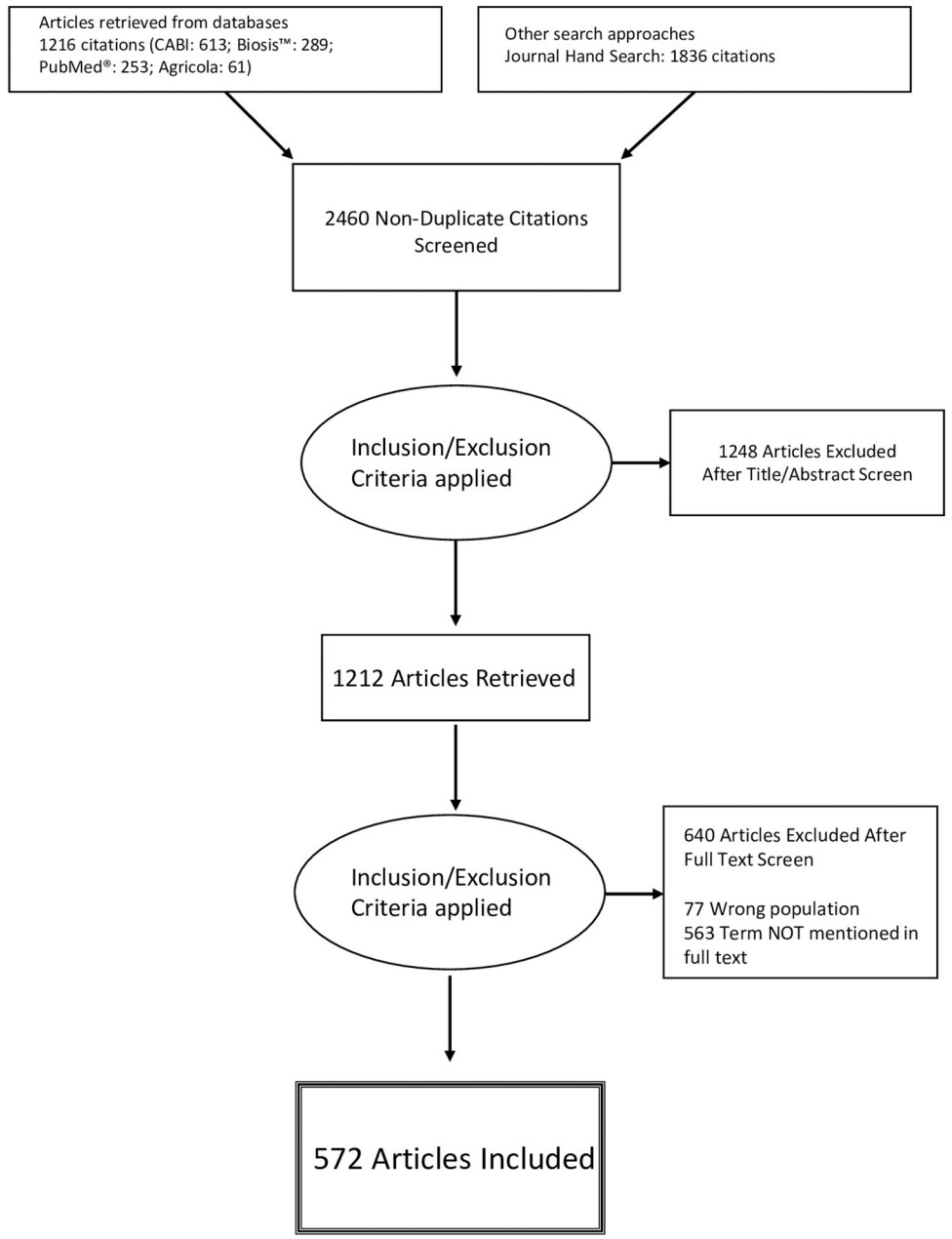

**Fig 1. PRISMA flowchart.** This flowchart depicts article inclusion for allostasis and allostatic load in non-human animal species.

corresponding to the number of publications in 2021 only represents half of the year. Species across all five main vertebrate groups, as well as invertebrates, were represented (Fig 3): invertebrates (n = 10), fish (n = 143), amphibians (n = 12), reptiles (n = 38), birds (n = 134), and mammals (n = 177). There were also publications that referred to animal populations, but not a specific species or taxa (n = 32) or included multiple taxa (n = 26). Most species were discussed in only one or two publications, while some species were well represented in multiple publications (S1 Appendix). The most commonly studied fish species were Atlantic salmon (*Salmo salar* L.) (n = 21), gilthead seabream (*Sparus aurata* L.) (n = 20), and rainbow trout

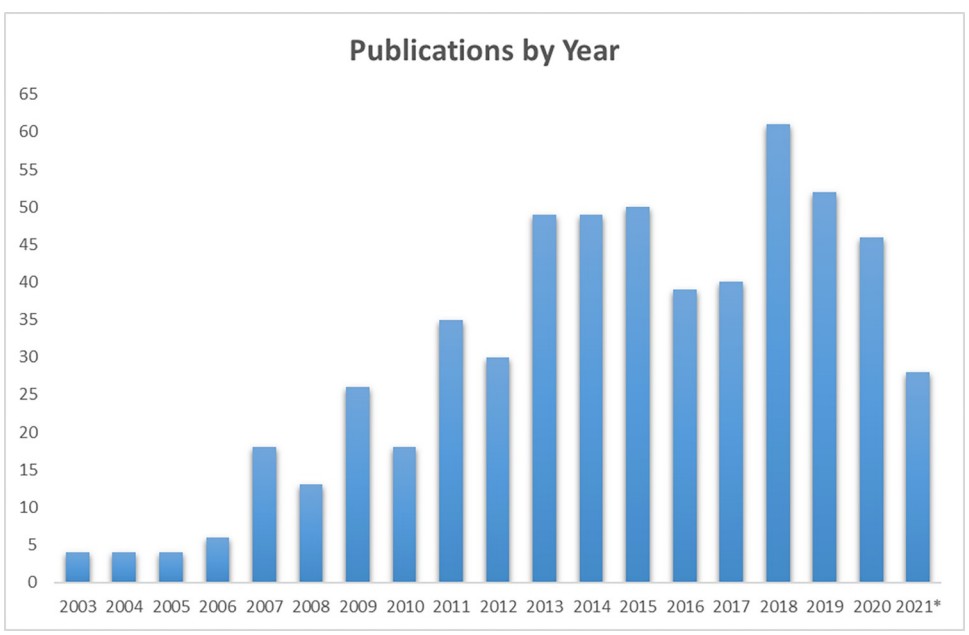

**Fig 2. Publications by year that used allostasis terminology in animals.** A total of 572 articles were identified in the non-human animal literature using terms related to allostasis and/or allostatic load. *Search results for 2021 are through June and only represent half of the publications from this year.

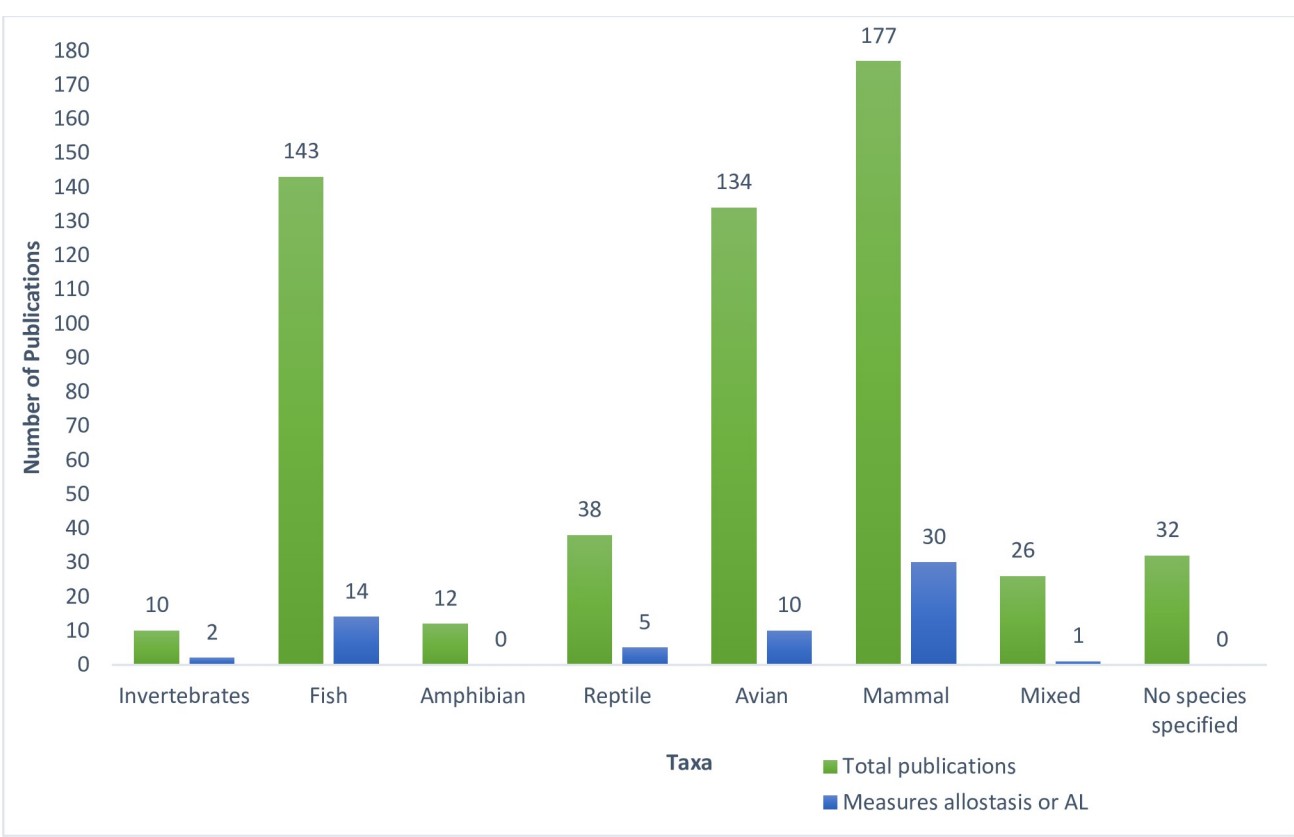

**Fig 3. Publications that used allostasis terminology and directly assessed allostatic load in animals broken down by taxa.** A total of 572 articles were identified in the non-human animal literature using terms related to allostasis and/or allostatic load across all taxa. Of these, 63 directly assessed and made conclusions about allostatic load.

(*Oncorhynchus mykiss*) (n = 11). The most commonly studied reptile species were common lizards (*Zootoca vivipara*), and Eastern fence lizards (*Sceloporus undulatus*) with 6 publications each. The most commonly studied bird species were house sparrows (*Passer domesticus*) (n = 15), zebra finches (*Taeniopygia guttata*) (n = 7), and chickens (*Gallus gallus domesticus*) (n = 6). The most studied mammalian species were cows (*Bos taurus*) (n = 11), rats (*Rattus* spp.) (n = 11), and rhesus macaques (*Macaca mulatta*) (n = 8). An array of invertebrates and amphibian species were represented, but none that were studied in more than two publications.

## Biomarkers and assessment of allostatic load

Of the 572 publications, a total of 63 (11%) measured biomarkers to make assessments of allostasis or AL, 61 primary research publications (Table 1) and two meta-analyses [45, 46]. The

**Table 1. Summary of publications that directly assessed allostatic load in animals.**

| Animal Class | Species | Biomarkers Evaluated | Major conclusions | ALI? |
|---|---|---|---|---|
| Invertebrates | Marine crab (*Hepatus pudibundus*) | Cl-<br>Lactate<br>Mg<br>Osmolality | • Lactate was a biomarker of AL in a 20% salinity after 6–12 hours [47] | No |
| | Marine/estuarine swimming crab (*Callinectes danae*) | Cl-<br>Lactate<br>Mg<br>Osmolality | • No biomarkers were indicative of AL in this species [47] | No |
| | Freshwater prawn (*Macrobrachium acanthurus*) | Cl-<br>Lactate<br>Mg<br>Osmolality | • Muscle dehydration was associated with AL in 30% salinity, but hemolymph biomarkers were not [47] | No |
| | Caribbean spiny lobster (*Panulirus argus*) | HR | • Artificial light at night did not represent a meaningful challenge to AL [48] | No |
| Fish | Gilthead seabream (*Sparus aurata* L.) | Cortisol<br>Glucose<br>Hct<br>Hg<br>Lactate | • Fish exposed to multiple sensory perception stressors had overall downregulation of mitochondrial-related genes, the greatest circulating cortisol and reduced lactate reflective of an increased AL [49] | No |
| | | Cortisol<br>Glucose<br>Lactate<br>Protein | • Decreased levels of mucus cortisol were indicative of reduction in AL due to effects of a phytogenic supplemented diet [40] | No |
| | | α-MSH<br>Cortisol | • There were associations between stress load and basal cortisol, but no impact on performance on an acute stress test<br>• Chronic stress had no effect on basal plasma α-MSH [50] | No |
| | Creek chub (*Semotilus atromaculatus*) | Cl-<br>Cortisol<br>Glucose<br>Glutathione<br>Hct<br>K+<br>Lactate<br>Na+ | • Creek chubs found in agricultural stream reaches maintained allostasis of physiological systems, despite a lower-magnitude cortisol and metabolic response [51] | No |
| | Rainbow trout (*Oncorhynchus mykiss*) | Cortisol<br>Free amino acids<br>Glucose<br>Growth hormone<br>Total protein | • Intermittent social contact between fish resulted in an increased AL in both dominant and subordinate individuals [52] | No |
| | | HR | • Netting a fish and placing it in a new social setting resulted in a greater AL than netting alone, indicated by an elevation of heart rate for the 8 hours following the event [53] | No |
| | Atlantic Salmon (*Salmo salar* L.) | Cortisol Na+ PCO2 pH | • Elevations in cortisol indicate an AL and increased stress associated with high stocking density [54] | No |
| | | Cortisol<br>Glucose | • Fish exposed to aluminum toxicity in acidic water had elevated cortisol and glucose and therefore experienced high AL [55] | No |
| | | Ca2+<br>Cl-<br>Cortisol<br>K+<br>Na+ | • No differences seen between treatment groups<br>• Episodic exposure to aluminum in water with a pH of 5.5 may not result in higher AL compared to control conditions or continuous exposure [56] | No |
| | | Cortisol | • Fish exposed to highest aluminum concentrations did not mount an appropriate cortisol response to additional stressors<br>• Indicators of AL were used to show impacts of water acidification [57] | No |
| | | Cl-<br>Cortisol<br>Mg | • Fish that were stressed prior to handling for vaccinations had elevated plasma cortisol<br>• Elevation of plasma cortisol prior to vaccination could result in AL<br>• Increased AL could have a significant impact on welfare [58] | No |
| | Alabama bass (*Micropterus henshalli*) | Cortisol Neutrophil: Lymphocyte ratio | • Fish that were exposed to regulated flows had high baseline cortisol reflective of an allostatic state [2] | No |
| | European seabass (*Dicentrarchus labax*) | α-MSH<br>Cortisol | • Stress load significantly affected basal cortisol and performance on an acute stress test<br>• Chronic stress had no effect on basal plasma α-MSH [50] | No |
| | | Cortisol | • There was no impact of tank volume on AL in larval sea bass [59] | No |
| | Cichlid (*Neolamprologus pulcher*) | Cortisol | • Dominant fish had higher circulating glucocorticoids compared to subordinate indicating a higher AL [60] | No |

(*Continued*)

**Table 1.** (Continued)

| Animal Class | Species | Biomarkers Evaluated | Major conclusions | ALI? |
|---|---|---|---|---|
| Reptiles | Eastern Fence lizard (*Sceloporus undulatus*) | Corticosterone | • Lizards exposed to invasive fire ants for longer periods of time had lower concentrations of corticosterone indicating AL [61] | No |
| | Colorado Checkered Whiptail (*Aspidoscelis neotesselata*) | BKA<br>Corticosterone<br>Estradiol<br>Free glycerol<br>Glucose<br>Triglycerides | • The patterns of hormones reflecting reproductive status, energy metabolism and innate immunity varied with season and vitellogenic stage<br>• Results partially supported the hypothesis that allostatic changes occur seasonally and throughout reproduction [62] | No |
| | Plateau side-blotched lizards (*Uta stansburiana uniformis*) | BKA<br>Corticosterone<br>Glucose | • Corticosterone and BKA increased during acute stress whereas glucose decreased demonstrating allostatic outcomes from acute stress [63] | No |
| | Common lizard (*Zootoca vivipara*) | Corticosterone Triglycerides | • Higher corticosterone was measured in both adults and yearlings in low-quality habitat<br>• AL may increase in degraded habitats [64] | No |
| | Pygmy rattlesnakes (*Sistrurus miliarius*) | Corticosterone | • Snakes with severe clinical signs of snake fungal disease had elevated corticosterone and therefore high AL [65] | No |
| Avian | Black grouse (*Tetrao tetrix*) | Corticosterone | • Fecal corticosterone and foraging patterns suggested that birds with high corticosterone concentrations are experiencing AL [14] | No |
| | Black kite (*Milvus migrans*) | Corticosterone | • Nestlings exposed to low temperatures had high corticosterone<br>• Corticosterone was negatively related to body condition score and brood hierarchy rank<br>• Results support the use of corticosterone to infer AL in this species [66] | No |
| | | Corticosterone | • Feather corticosterone levels were highest in young birds and declined with age<br>• Feather corticosterone levels were negatively associated with body size<br>• Feather corticosterone provides an indication of how AL vary over the life of an individual [67] | No |
| | Black-legged kittiwake (*Rissa tridactyla*) | Corticosterone | • The relationship between AL, which was measured using corticosterone values, and energy stores was curvilinear at both life history stages [68] | No |
| | Blue fronted Amazon Parrots (*Amazona aestiva*) | Glucocorticoids | • Dominant birds had higher AL than subordinates, indicated by higher fecal glucocorticoid metabolites [69] | No |
| | European starlings (*Sturnus vulgaris*) | Corticosterone HR | • Decreased temperatures and exposure to rain caused elevated heart rates and corticosterone levels in non-molting starling indicating that these birds had increased AL [70] | No |
| | Griffon vulture (*Gyps fulvus*) | Cortisol DHEA | • Cortisol and DHEA levels were higher in the feathers of physiologically compromised birds compared to the healthy control group<br>• Moulted feathers can be a non-invasive means of evaluating AL in birds [71] | No |
| | Mallard ducks (*Anas platyrhynchos*) | Corticosterone | • Feather corticosterone reflected energy expenditure and was therefore a proxy of AL<br>• Feather corticosterone provided retrospective information about AL early in life that was not detected from the body condition of birds in real time [72] | No |
| | Red crossbill (*Loxia curvirostra*) | Corticosterone<br>Corticosterone binding globulin capacity<br>Testosterone | • No association was seen between parasite load and corticosterone levels<br>• Conclude that parasite infections did not affect AL in this species [73] | No |
| | Red kites (*Milvus milvus*) | Corticosterone DHEA | • Free-living kites had higher levels of corticosterone than captive birds, indicating a higher AL [74] | No |
| Mammals | Assamese macaques (*Macaca assamensis*) | Glucocorticoids | • During the mating season females that were in closer association with males had lower fecal glucocorticoids<br>• In the non-mating season positive female-female socializing resulted in lower fecal glucocorticoids<br>• Positive social interactions resulted in lower AL based on glucocorticoid measures [75] | No |
| | Barrow Island euro (*Macropus robustus isabellinus*) | Cortisol<br>Hb<br>Leukocyte count<br>LVP<br>Osmolality<br>PCV<br>RBC<br>Reticulocyte count | • When exposed to prolonged drought animals experienced AL, but were able to maintain their normal homeostatic state<br>• Conclusions were based on elevated osmolality, cortisol and LVP [76] | No |
| | Black capuchins (*Sapajus nigritus*) | Glucocorticoids | • Decrease fruit intake during the dry season resulted in increased fecal glucocorticoids in juvenile males<br>• Glucocorticoid levels in adult males was more impacted by breeding season<br>• Fruit consumption as the main source of AL for immature animals, while reproductive costs had greater effect on adults [77] | No |
| | Blue monkey (*Cercopithecus mitis*) | Glucocorticoids | • Directly equated AL with deviation of fecal glucocorticoids from the baseline • Participating in social play may decrease social uncertainty and AL [78] | No |
| | Brush-tailed bettong (*Bettongia penicillata*) | Glucocorticoids | • Fecal glucocorticoids were not elevated after wildfires indicating that bettong maintain allostasis in the period immediately following a fire [79] | No |
| | Cairo spiny mouse (*Acomys cahirinus*) | Glucocorticoids | • There is a significant interaction between parasitism and social contact<br>• Solitary females with no parasitism had the highest glucocorticoid levels<br>• Social contact and parasite infestation may lessen AL in pregnant rodents [80] | No |
| | Capuchin monkeys (*Sapajus libidinosus*) | Cortisol<br>Testosterone | • Dominant males had higher basal and peak fecal cortisol levels, and therefore had higher AL than subordinates [81] | No |
| | Cheetah (*Acinonyx jubatus*) | Glucocorticoids | • No difference in fecal glucocorticoid levels between parous and nulliparous females<br>• AL may not impact reproductive success [82] | No |
| | | Cortisol | • Cheetahs had lower serum cortisol concentrations than leopards<br>• Capture may cause a higher AL in leopards [83] | No |
| | Chimpanzees (*Pan troglodytes*) | Cortisol | • Anesthesia for routine examinations resulted in increased urinary cortisol<br>• Findings indicate a major disruption of homeostasis and an AL [84] | No |
| | Dog (*Canis familiaris*) | Cortisol<br>DHEA-S | • Maternity plays a role in HPA axis activation resulting in chronic cortisol secretion leading to increased AL [85] | No |

(*Continued*)

**Table 1.** (Continued)

| Animal Class | Species | Biomarkers Evaluated | Major conclusions | ALI? |
|---|---|---|---|---|
| | Gorilla (*Gorilla gorilla gorilla*) | Albumin Cortisol CRH DHEA-S Glucose IL-6 TNF-α | • AL had a positive association with age and total stress events • AL was higher in females than males • AL had no association with parity in females [86] | Yes |
| | | Albumin Cortisol CRH DHEA-S Glucose IL-6 TNF-α | • Sex and rearing history impact AL • Females had higher AL than males • Wild-caught females had significantly higher AL than mother-reared gorillas [87] | Yes |
| | | Albumin Cortisol CRH DHEA-S Glucose IL-6 TNF-α | • With expanded sample size associations between AL and sex, age, stress events and rearing history remained [88] | Yes |
| | | Albumin Cholesterol Cortisol CRH DHEA-S HOMA-IR IL-6 TNF-α Triglycerides | • Adding total cholesterol and triglycerides into the ALI improved prediction of morbidity, cardiac disease and mortality in zoo-housed animals [89] | Yes |
| | Grey mouse lemur (*Microcebus murinus*) | Glucocorticoids | • Fecal glucocorticoids were higher in the dry seasons • Lemurs may experience higher AL during dry season [90] | No |
| | Leopard (*Panthera pardus*) | Cortisol | • Cheetahs had lower serum cortisol concentrations than leopards • Capture may cause a higher AL in leopards [83] | No |
| | Mandrill (*Mandrillus sphinx*) | Glucocorticoids | • There was no relationship between dominance rank and glucocorticoid levels • Suggests no difference in AL between dominant and subordinate individuals [91] | No |
| | Pig (*Sus scrofa domesticus*) | Cortisol DHEA-S | • Levels of cortisol and DHEA-S in pig hair differed between two different farms showing that they had different AL [92] | No |
| | Plains zebras (*Equus quagga*) | Glucocorticoids | •Animals in large aggregations had higher fecal glucocorticoid levels than those in medium or small aggregations • Migratory zebras may have higher AL in large aggregations [93] | No |
| | Przewalski's horses (*Equus ferus przewalskii*) | HR HRV | • Noted a drop in HRV with a peak in HR in the spring indicative of AL associated with increased energy demands [19] | No |
| | Rat (*Rattus norvegicus domestica*) | Corticosterone | • Compared to rats that were chronically restrained those that had chronic variable stress had higher basal corticosterone and therefore higher AL [94] | No |
| | | Cortisol Creatine Glucose IL-1β IL-2 IL-6 Lactate Leptin Weight | • Operationalized multiple biomarkers to create a rat cumulative allostatic load model (rCALM) which estimated the burden of chronic stress and indicated future disease risks • Individual biomarkers did not accurately reflect neuronal deficits whereas AL did [95] | Yes |
| | Red deer (*Cervus elaphus*) | Cortisol | • HCC was higher in areas of higher deer density, suggesting that AL is higher in areas of higher density and harder environmental conditions [96] | No |
| | | Cortisol | • Cortisol levels varied based on sampling area indicating differences in AL [97] | No |
| | Rhesus Macaques (*Macaca mulatta*) | IL-1ra IL-6 IL-8 Glucocorticoids | • Older females had higher IL-1ra concentrations than younger females • Females had higher glucocorticoid levels when pregnant and lactating • Findings suggest that some individuals experience higher AL than others [41] | No |
| | Sheep (*Ovus* spp) | ACTH Cortisol Temperature | • Sheep that were chronically stressed by individual housing and sleep deprivation showed HPA-axis dysregulation, suggesting an increased AL in the chronically stressed group of animals [98] | No |
| | Spotted hyena (*Crocuta crocuta*) | Glucocorticoids | • Variations in fecal glucocorticoids based on dominance provide evidence that hunger and sibling competition affect AL in spotted hyenas [99] | No |
| | | Glucocorticoids | • During breeding periods when there was interaction between male competitors, the low-ranking males had higher fecal glucocorticoids than high- ranking males • There was no difference in fecal glucocorticoids between the groups when individuals were alone or competitors were absent • Differences in AL across male social ranks may be a result of interactions with other males [100] | No |
| | White faced capuchins (*Cebus capucinus*) | Dihydro-testosterone Glucocorticoids Testosterone | • Compared to subordinate males the dominant males have higher fecal testosterone, dihydrotestosterone and fecal glucocorticoids • Regardless of dominance status all males had elevated fecal glucocorticoids in the presence of fertile females • Findings suggest that there is a cost of dominance, but that in the presence of fertile females AL increases for all males [101] | No |
| | Yellow-bellied marmots (*Marmota flaviventris*) | Glucocorticoids | • Young marmots had lower fecal glucocorticoids in rural environments, therefore lower AL • Adult marmots had higher fecal glucocorticoids in rural environments • All age groups exhibited parabolic relationships between degree of urbanization and fecal glucocorticoids [102] | No |

A total of 61 primary research publications that made direct assessments about allostasis and/or allostatic load in non-human animals, including animal class, species, biomarkers evaluated, major conclusions and whether an allostatic load index (ALI) was constructed.

Abbreviations: ACTH, adrenocorticotropic hormone; AL, allostatic load; ALI, allostatic load index; α-MSH, α-melanocyte–stimulating hormone; BKA, bacterial killing activity; Ca2+, calcium; Cl-, chloride; CRH, corticotropin-releasing hormone, DHEA, dehydroepiandrosterone; DHEA-S, dehydroepiandrosterone-sulfate; Hb, hemoglobin; HCC, hair cortisol concentration; Hct, hematocrit; HOMA-IR, homeostatic model assessment of insulin resistance; HR, heart rate; HRV, heart rate variability; IL-1ra, interleukin-1 receptor antagonist; IL-2, interleukin-2; IL-6, interleukin-6; IL-8, interleukin-8; K+, potassium; LVP, lys8-vasopressin; Mg, magnesium; Na+, sodium; PCO2, partial pressure of carbon dioxide; PCV, packed cell volume; RBC, red blood cells; TNF-α, tumor necrosis factor alpha

two meta-analyses are not included in Table 1 due to the multiple species evaluated and the lack of primary data.

These 63 papers represented all taxa except amphibians (Fig 3) and numerous biomarkers were evaluated (Table 2). A vast majority of the publications (58/63; 92%) measured glucocorticoids (cortisol: n = 28, glucocorticoid metabolites: n = 16, corticosterone: n = 14) and in 25 papers (43%), glucocorticoids were the sole biomarker measured. Depending on the study, glucocorticoids were measured in a variety of tissues, including hair, feces, urine, plasma, and feathers.

Of the 63 publications that directly assessed allostasis or AL, only 6 (9.5%), many of which were from our group, constructed an ALI using published methodology [45, 86, 87, 89, 95].

## Discussion

The overall goal of this scoping review was to evaluate the extent to which principles of allostasis and AL are being applied to non-human animals. Of the 572 articles included in this review, most were written within the last ten years. Over the last 5 years, there have been 40–60 peer-reviewed publications annually. This change reflects a growing application of allostasis and AL in animal populations.

The species diversity encompassed all taxonomic groups, with over 250 different species represented across the 572 articles. Seventy-nine percent of the publications discussed mammalian, fish, or avian species, with far fewer papers focusing on reptiles, amphibians or invertebrates. This gap in the literature presents opportunities for future research, particularly in taxa

**Table 2. Biomarkers used to make direct assessments about allostatic load in animals.**

| Biomarker | Number of publications |
|---|---|
| Cortisol | 28 |
| Glucocorticoid metabolites | 16 |
| Corticosterone | 14 |
| Glucose | 11 |
| DHEA-S | 6 |
| IL-6 | |
| Lactate | 5 |
| Albumin | 4 |
| Cl- | |
| CRH | |
| HR | |
| TNF-α | |
| Na+ | 3 |
| Testosterone | |
| Triglycerides | |
| BKA | 2 |
| DHEA | |
| Hb | |
| Mg | |
| K+ | |
| Osmolality | |
| PCV | |
| Total protein | |

(*Continued*)

**Table 2.** (Continued)

| Biomarker | Number of publications |
|---|---|
| α-MSH | 1 |
| ACTH | |
| Ca2+ | |
| Cholesterol | |
| Corticosterone binding globulin capacity | |
| Creatinine | |
| Dihydrotestosterone | |
| Estradiol | |
| Free amino acids | |
| Free glycerol | |
| Glutathione | |
| Growth hormone | |
| HRV | |
| HOMA-IR | |
| IL-1β | |
| IL-1ra | |
| IL-2 | |
| IL-8 | |
| Leptin | |
| Leukocyte count | |
| Locomotion | |
| LVP | |
| Monoamines | |
| Neutrophil: Lymphocyte ratio | |
| PCO2 | |
| pH | |
| RBC | |
| Reticulocyte count | |
| Temperature | |
| Weight | |

This table includes biomarkers used in 61 primary research publications to make direct assessments about allostasis and/or allostatic load in non-human animals.

Abbreviations: ACTH, adrenocorticotropic hormone; α-MSH, α-melanocyte–stimulating hormone; BKA, bacterial killing activity; Ca2+, calcium; Cl-, chloride; CRH, corticotropin-releasing hormone, DHEA, dehydroepiandrosterone; DHEA-S, dehydroepiandrosterone-sulfate; Hb, hemoglobin; Hct, hematocrit; HOMA-IR, homeostatic model assessment of insulin resistance; HR, heart rate; HRV, heart rate variability; IL-1β, interleukin-1 beta; IL-1ra, interleukin-1 receptor antagonist; IL-2, interleukin-2; IL-6, interleukin-6; IL-8, interleukin-8; K+, potassium; LVP, lys8-vasopressin; Mg, magnesium; Na+, sodium; PCO2, partial pressure of carbon dioxide; PCV, packed cell volume; RBC, red blood cells; TNF-α, tumor necrosis factor alpha

like amphibians, which are facing global population declines due to diseases like *Batrachochytrium dendrobatidis* (Bd) [103]. Since many disease states are diagnosed and/or monitored by measuring biomarkers (e.g., cholesterol and insulin resistance for metabolic syndrome in humans), ALIs encompassing multiple somatic systems have the potential to predict future health outcomes. The link between stress and risk of disease is well described in a wide variety of species, and ALIs have been used to characterize disease risks in human populations [104, 105]. Additionally, ALIs have the potential to be used to assess the impact of social and

environmental stressors and how they drive animal movements and affect disease ecology on a broader scale [106].

Even within the largely represented groups, like fish, there is potential for expanded applications of AL. Many of the fish publications focused on the health and welfare of commercial species in aquaculture settings [54, 58, 107], with 61 publications looking at Atlantic salmon, gilthead seabream, rainbow trout or European seabass (*Dicentrarchus labax*). In contrast, only three papers explored allostasis in zebra fish (*Danio rerio*) [108–110], which are an important laboratory species that may benefit from studies of AL.

Despite 572 publications mentioning allostasis or AL, only a small proportion (63 publications, 11%) made direct assessments using physiological biomarkers. The other 509 articles only mentioned AL hypothetically as part of the introduction or discussion of other findings. Of this subset of 63 articles, 61 were primary research studies and two were meta-analyses that incorporated data from multiple publications to make their assessment. Within the primary research, 49 different species were studied across all taxa, with the exception of amphibians.

There were common themes amongst the 63 publications, including a focus on environmental challenges, social structure, and animals under managed care. Evaluating AL in the context of environmental challenges was a focus of several publications and encompassed different focus areas including the impacts of human activities (i.e., agriculture [51], urbanization [102], snow sports [14], and pollution [74]), and characterization of the effects of environmental parameters on AL (i.e., weather [70], fire [79], and salinity [47]). Several publications investigated the effects of social structure on AL in several species including hyena [100], Assamese macaques [75], bearded capuchins [81], cichlids [60], and rainbow trout [52]. Multiple researchers aimed to investigate the connection between animal management techniques and AL [53, 95], in some cases with an explicit emphasis on animal welfare [54, 92].

While each of the 63 publications drew direct conclusions about AL from their findings, the data need to be interpreted with caution. One of the biggest methodological challenges was that a high proportion (44%) of studies made conclusions about allostasis or AL based on glucocorticoids alone. This finding is unsurprising, as glucocorticoids have historically been considered the principle hormonal mediator for AL [26, 111, 112]. However, there are limitations to using glucocorticoids as the only measure of stress [113, 114]. For example, substantial inter- and intra-individual variation, as well as fluctuations due to temporary coping mechanisms associated with season or reproductive effort, complicate their interpretation [115–118]. Moreover, research in humans indicates that individual biomarkers are inadequate for estimating AL [119–123]. Therefore, given the advances in methodology, conclusions about AL cannot be made based on glucocorticoids alone.

A second consideration in interpreting the findings of many of these 63 publications is the use of AL to evaluate response to an acute stressor, often using a single-biomarker like glucocorticoids (e.g., [53, 70, 71, 85]). This deviates from the original purpose for the development of ALIs in humans, which was to estimate wear-and-tear as the result of chronic, long-term stress. Even when estimated using multiple biomarkers, AL should not be used to replace glucocorticoids as a measure of an acute stress response. For instance, Arlettaz et al. (2015) characterized the impact of free-riding snow sports had on black grouse (*Tetrao tetrix*) in an alpine habitat. To mimic the disturbance of sports, grouse were flushed on consecutive days and subsequently had elevated fecal glucocorticoids compared to baseline. Based on these findings it was concluded that repeated disturbances resulted in an increased AL and thereby presented a threat to wildlife populations. Similarly, Hing et al. (2016) investigated the effects of wildfires on brush-tailed bettongs (*Bettongia penicillata*) by measuring fecal glucocorticoids two days after a fire. When there was no significant elevation of glucocorticoids compared to baseline, it was concluded that this species adapts to these environmental challenges with no effect on AL.

While these types of evaluations are essential to monitor high-risk populations, neither study adequately assessed AL, as the conclusions were based only on changes in cortisol levels occurring over a short time period.

Although ALIs have traditionally been used to evaluate the impact of chronic, long-term stressors in humans, there are challenges with this approach in non-human animal species, particularly wildlife, as it can be difficult to disentangle acute stress from chronic stress. For example, animals must be manually or chemically restrained to obtain a blood sample, which likely increases some of the biomarkers of interest, such as cortisol and glucose [124, 125]. Additionally, repeated sampling of animals can be challenging in wild settings, making it difficult to identify all the potential stressors an animal encounters over time. While it would not be an estimate of AL researchers may be able to adapt the methodology to try and consider using an index of biomarkers that would be expected to increase in the face of acute, short-term stress, such as glucose, cortisol, and catecholamines. This approach can allow us to gain a more robust understanding of the "cost" associated with acute stress in animals compared with single biomarkers such as cortisol [126].

Only six of the 63 publications that made direct assessments about AL used an ALI following the original method used in human research [31]. Western lowland gorillas (*Gorilla gorilla gorilla*) were the only non-human primate species in which an ALI was constructed and used to evaluate AL in four articles by the same research group [86–89]. Seven biomarkers were measured for the gorilla ALI, including albumin, cortisol, corticotropin releasing hormone (CRH), DHEA-S, glucose, IL-6, and tumor-necrosis factor (TNF)-α. Older animals, males, and gorillas with a higher number of stressful events over their lifetime had higher AL [86]. In a follow-up study, the authors found that wild-caught female gorillas had higher AL than mother-reared gorillas, although there was no difference in AL by rearing history for males [87]. Building upon their initial model, the authors found that the associations of AL with sex, age, stressful events, and rearing history remained when additional institutions were incorporated [88]. The authors later expanded the ALI and found that adding cholesterol and triglycerides improved predictions of morbidity and mortality risk in zoo housed gorillas [89].

In another publication that constructed an ALI, the authors proposed a measure of AL that they referred to as the "rat cumulative allostatic load measure (rCALM)" [95]. This study included the following biomarkers: cortisol, blood glucose, body weight, interleukin-1β (IL-1β), interleukin-2 (IL-2), IL-6, leptin, lactate, and creatine. The authors found that when evaluated individually the biomarkers were not predictive for neuronal deficits. However, when used as a comprehensive ALI, rCALM was an effective predictor of neurologic deficits. The authors concluded that the rCALM index estimated the effects of chronic stress and could potentially be used to indicate future disease risks.

The last publication that constructed an ALI used a meta-analysis. The aim of this publication was to evaluate hypotheses that explain variation in parasitism based on social status in vertebrate species [45]. The authors combined data from multiple studies and determined AL based on previously described methodology [46]. Authors concluded that AL was not correlated with relative parasitism in vertebrates [45]. However, the Goymann and Wingfield (2004) method used for calculating AL in this study deviates from the standards used in human populations. Instead of determining AL using an ALI comprised of multiple biomarkers, AL scores were assigned to individuals based on the assumed costs of becoming dominant within a social group; assessments were then made based on each individual's assigned AL [46]. These assumed costs of dominance were based on cortisol levels in dominant vs. subordinate animals. However, cortisol alone is an insufficient proxy for AL, making this methodology problematic. Future meta-analyses that incorporate biomarkers measured in a species across multiple publications are encouraged.

Many of the papers reviewed here, both those that made direct assessments about AL and those that did not, measured multiple biomarkers, but assessed them individually and not as an index (e.g., [41, 49, 127]). Thus, there is an opportunity to re-assess previously collected data as an ALI, which may increase the power and impact of the data. For instance, Hudson et al (2020) measured six different biomarkers in Colorado checkered whiptail (*Aspidoscelis neotesselata*) that reflected reproductive status, energy metabolism and innate immunity. The authors assessed biomarkers individually to determine if there were AL changes based on season and reproductive status [62]. Instead, authors could have used these biomarkers to create an ALI to assess AL.

Since there is biological variation between taxa, there is likely not one single set of biomarkers that universally applies to all species, although it may be possible to determine a single ALI for a group of closely related species (e.g., great apes). Instead, we recommend that species-specific ALIs be constructed using biomarkers that are most reflective of long-term stressors. Biomarker discovery and advances in animal endocrinology are required to identify sufficient biomarkers to construct ALIs in many species. For example, we recently published a study that constructed an ALI to measure AL in ring-tailed lemurs (*Lemur catta*), and one of the largest challenges was finding biomarkers that could be measured in lemur serum with valid results [128]. In our case, several inflammatory cytokines were investigated as potential biomarkers for incorporation in the ALI but could not be reliably measured using the commercially available assays. It is important to note that one of the limitations of incorporating new biomarkers in an ALI is the lack of information about normal reference ranges in many species. Even when reference ranges are available, measured concentrations may be affected by the chemical restraint necessary for sample collection. However, researchers often uses sample-based cut-points instead of normal reference ranges to calculate AL; thus, an absence of reference ranges does not necessarily preclude the use of a biomarker for this type of research.

Future research using non-human animals should adopt a new way of thinking when assessing allostasis and AL. First, we encourage researchers to refrain from making conclusions about allostasis using glucocorticoids alone, and to focus on chronic rather than acute stressors. Second, we suggest that an ALI be constructed for each species using multiple relevant biomarkers that ideally reflect neuroendocrine, cardiovascular, immune, and metabolic systems as originally proposed for humans by Seeman et al. (1997). Using this approach, AL may become a useful measurement of stress and animal welfare. Indeed, several authors reflected on the importance of continuing to develop these measures as means of potentially evaluating chronic stress in animal populations [29, 82, 129] and acknowledged that AL may be an important tool in assessing animal welfare [43, 130]. The next step is to continue to refine approaches and methodologies to create practical and appropriate ALIs across species.

There are some limitations to the generalizability of this review. We only included manuscripts published in English, which excluded potentially relevant literature published in other languages. We also excluded studies that used laboratory rodents as models for humans, which may have limited the number of studies using common laboratory animals such as rodents and primates. Finally, "invertebrates" was not used as a specific search term; however, several invertebrate publications were found in the search results and included in the review. As a result, this review likely underestimated the number of papers that are applying allostasis or allostatic load to invertebrate species.

## Conclusion

This review describes how principles of allostasis and allostatic load are being used in research with non-human animals. We identified a total of 572 peer-reviewed publications that

mentioned allostasis and/or allostatic load published since 2003, covering a variety of non-human animal species. Of these, 63 made direct assessments about allostatic load in animals. However, many of these assessments were based on single biomarkers, such as glucocorticoids, and were focused on the effects of acute rather than chronic stressors. Future research in animals is encouraged in this area, with an emphasis on the creation of allostatic load indexes. Researchers should also use more consistent methodologies when assessing allostatic load, such as those already established in human research.

## Supporting information

**S1 Checklist. Preferred Reporting Items for Systematic reviews and Meta-Analyses extension for Scoping Reviews (PRISMA-ScR) checklist.**
(DOCX)

**S1 Dataset. Full reference list of 572 articles included in the scoping review.**
(DOCX)

**S1 Appendix. Taxonomic and species breakdown of literature applying the concepts of allostasis and allostatic load to animals.**
(DOCX)

## Author Contributions

**Conceptualization:** Kathryn E. Seeley, Kathryn L. Proudfoot, Ashley N. Edes.

**Data curation:** Kathryn E. Seeley.

**Methodology:** Kathryn E. Seeley, Kathryn L. Proudfoot.

**Supervision:** Kathryn L. Proudfoot, Ashley N. Edes.

**Writing – original draft:** Kathryn E. Seeley.

**Writing – review & editing:** Kathryn L. Proudfoot, Ashley N. Edes.

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
