## [Decision Letter · Decision Letter 0]

4 Jul 2022

PONE-D-22-08746The application of allostasis and allostatic load in animal species: A scoping reviewPLOS ONE

Dear Dr. Seeley,

Thank you for submitting your manuscript to PLOS ONE. After careful consideration, we feel that it has merit but does not fully meet PLOS ONE’s publication criteria as it currently stands. Therefore, we invite you to submit a revised version of the manuscript that addresses the points raised during the review process. Two experts in the field have evaluated your study, and both reviewers found it relevant for publication, but raised some issues, althought minor, that need to be addressed before full consideration for publication. You will find their comments and suggestions below.

We look forward to receiving your revised manuscript.

Kind regards,

Sylvain Giroud, PhD

Academic Editor

PLOS ONE

Journal Requirements:

Additional Editor Comments:

Please take care of the reviewers' comments and make the necessary changes to the manuscript.

Reviewers' comments:

Reviewer's Responses to Questions

**Comments to the Author**

1. Is the manuscript technically sound, and do the data support the conclusions?

Reviewer #1: Yes

Reviewer #2: Yes

Reviewer #3: Yes

2. Has the statistical analysis been performed appropriately and rigorously? 

Reviewer #1: Yes

Reviewer #2: N/A

Reviewer #3: Yes

3. Have the authors made all data underlying the findings in their manuscript fully available?

Reviewer #1: Yes

Reviewer #2: Yes

Reviewer #3: Yes

4. Is the manuscript presented in an intelligible fashion and written in standard English?

Reviewer #1: Yes

Reviewer #2: Yes

Reviewer #3: Yes

5. Review Comments to the Author

Reviewer #1: In the presented study the authors reviewed the scientific literature to describe the extent to which the concepts of allostasis and allostatic load are mentioned as well as being applied either directly or indirectly to non-human animal taxa and species. Further, it was assessed which species and contexts are represented, which biomarkers are being used, and where an allostatic load index was constructed. Overall this study is giving a highly useful overview of the currently available scientific literature. Based on the outcome the authors discuss gaps in the literature and areas for future research and give recommendations for the application of the concept in animals within the context of chronic stress.

Strength:

This is an easy to read, comprehensive and clear review study on a relevant concept with increasing importance for several scientific disciplines. The methods applied are adequate and transparent. Further, in line with other scientists, the authors highlight the limitations in using glucocorticoids or glucocorticoid metabolites as a single measure to assess allostatic load and recommend the inclusion of several biomarkers to evaluate chronic stress conditions in vertebrates. This is one of the primary “methodological” aspects where a paradigm shift is essential to better understand the phenomenon of stress and its associated costs and consequences.

Limitations:

As the ALI, in my humbled opinion, is and will be an important concept for future research, it should be the goal to address a broad scientific audience. However (despite the wide range of taxa where AL was al lease mentioned mentioned in the leterature), many scientists, even if working in closely related fields, may not be aware of how to apply AL/ALI in a meaningful manner in different contexts. Therefor I encourage the authors to elaborate on certain aspects to provide the links between allostasis/allostatic load and e.g. disease susceptibility, epidemiology, reproduction, etc. (for details please see my comments below).

Line 58:

Please add “short-term” to acute stress (as done for chronic with “prolonged” “or long-term”.

I am aware this is an old discussion but I think the time aspect is important.

Line 64 and 65:

concerning “leukocyte values”: do the others refer here to shifts in leukocyte numbers and composition (according to the citation)? Please clarify. The authors might also add leukocyte function (e.g. Huber, N., Marasco, V., Painer, J., Vetter, S. G., Göritz, F., Kaczensky, P., & Walzer, C. (2019). Leukocyte coping capacity: an integrative parameter for wildlife welfare within conservation interventions. Frontiers in veterinary science, 6, 105.)

Line 66:

…..“there is no one measure that”….;

only a suggestion: there is no single measure….

Line 79:

….“allowing for a more flexible adaptation to stressors”.

I me be mistaken here but in my view allostasis and the two mentioned major axes allow to adapt to/cope with stressors in general (not “more” flexible). Further, stressors may not only specifically be coming from the environment but also from e.g. disrupted internal conditions independent from the environment). Apologies for being picky here.

Line 93:

I do not really see how the listed parameters mentioned in the next few lines reflect immune function. I suggest skipping “immune” here.

Line 133:

Please add a ® behind Pubmed. Please also check for the other databases if this is required (same for the databases in the Prisma flow diagram)

Lines 274, 275:

This is an important aspect pointed out here! However, for the reader (who may not be as familiar with stress physiology and the link between the neuroendocrine and the immune system and e.g. changes in movement behaviour and how this is linked to epidemiology etc.) it may not be clear how AL and diseases ecology are linked. Please add a brief explanation here.

Lines 302, 303.

“Thank you” for stressing this point!

Lines 309 to 311:

I absolutely agree, that using a single biomarker to assess AL is not sufficient and leads to problematic interpretations.

However: i) in my experience it is difficult to distinguish or entangle acute/short-term and chronic/long-term stress – especially when working with wildlife or animals which can only be sampled once and ii) does the initial purpose of AL and ALI to evaluate chronic stress conditions automatically invalidate its use to assess the “cost” linked to defined (known) acute stress reactions such capture or handling?

It would be helpful for the reader, if the authors could be more specific here in terms of either giving explanations to why AL/ALI should only be used in the context of its initial aim or consider that depending on the choice of biomarkers* and their interpretation AL/ALI could also be a suitable approach to assess acute stress reactions and the associated costs.

* Gormally, B. M., & Romero, L. M. (2020). What are you actually measuring? A review of techniques that integrate the stress response on distinct time‐scales. Functional Ecology, 34(10), 2030-2044.

Lines 372 to 373:

I go in line with authors on the suggested next and needed steps of biomarker discovery and advances in wildlife endocrinology. I do, however, also suggest to mention the gap of missing reference values for many wildlife species. In this context, it would also be nice if the authors mention that in many species it is only possible to obtain the necessary samples after capture and often immobilisation/anaesthesia, biasing the measured biomarkers and making their interpretation or use for AL difficult.

Lines 381 to 382:

Again, I absolutely agree – but please elaborate why! researches should focus on chronic stress (maybe you could do so in the discussion above (please see comment to lines 309 to 311).

Lines 384 to 386:

Although I personally are in line with the authors, again, the statement could be better justified in the discussion above, (link AL/ALI to welfare, which is straighter forward to understand – but please elaborate on the link to disease and why ALIs could be used as a predictor).

Line 403:

In accordance to my previous comment on the discussion of AL/ALI in the context of acute and chronic stress. Why do the authors see this as problematic?

Reviewer #2: In general, the manuscript is well-written and appears thorough.

The following are minor errors:

Line 138: Change "complimentary" to "complementary" (as in "complementary angle").

Line 141: Change "to being" to "to be".

Line 157: Change "thesis" to "theses".

Line 161: Change "Any article" to "Any articles" or change "were" to "was".

Line 293: The sentence "Several papers... trout [51]." appears somewhat incom

Line 299: Change from "data needs" to "data need".

Line 350: Change "Authors" to "The authors".

Reviewer #3: The authors performed a scoping review of allostatic load (AL) and its use in developing indices of allostatic load by reviewing studies in several animal species. Their conclusions should provide valuable information for others developing similar indices and, in our opinion, be applied to for the formation of a human AL index.

The manuscript missed some details that have indicated below. Also, there are multiple typos in the manuscript - it seems the authors did not do proper proofreading before submission.

Overall, we recommend acceptance for publication if they address these issues.

Keywords: allostasis; Allostatic load; allostatic load index; animals; Stress - change capital words

Line 96: waist–hip ratio, and glycosylated hemoglobin (HbA1c). Please add comma after waist-hip ratio to be consistent with your comma style.

LINE 118: The authors should briefly cite the tools used to conduct the scoping review (e.g. PRISMA, Arksey and O’Malley’s original framework)

Line 132: Have you used any reference management software? Please include.

Line 143-147: Please include semicolons to divide the journals included in the hand-search. The semicolon will help readers to keep track of the journal`s names.

Line 153: “An initial screening of the title and abstract was conducted”. Please add the number of authors that participated in each screening step and their initials.

Line 172: correct heartrate to heart rate throughout the paper.

Line 218: Fig 2. Please add * before “Search results for 2021 were through June and so only represent half of the publications from this year.”

Line 221: Fig 3. Please include the numbers found for each taxa in the figure legend.

Table 1: revise sentence: “No hemolymph biomarkers were but muscle dehydration was

associated with AL in salininity of 30%.” – Freshwater prawn (Macrobrachium acanthurus).

“Fish exposed to highest aluminum concentrations didn’t mount an appropriate cortisol ..” - Atlantic Salmon (Salmo salar L.). Contraction words should not be used.

Line 294: Please revise spaces between the text and reference as : “hyena[99], Assamese macaques [74], bearded capuchins [80], cichlids [59] and rainbow trout [51]..” Also, please check when the first letter should be capitalized or not as in Assamese macaques.

Revise scientific English and punctuation throughout the paper. (e.g. “Similarly, Hing et al investigated the effects that wildfires have on brush-tailed bettongs (Bettongia penicillata), by measuring fecal glucocorticoids two days after a fire.”

Line 324: Be consistent in the abbreviations: AL - allostatic load “put conservation mitigation actions into place, neither study adequately assessed allostatic load”

The reader could wonder what the actions/roles of each AL mediator are. I suggest creating a table describing the category each mediator fits. For instance: cytokines: immune system, pro- or anti-inflammatory actions, etc.

Line 342: Please add the Greek symbol in all interleukin-1b (IL-1b) references in the paper.

Line 346: Missing word. Revise sentence: “concluded that the rCALM index estimated the effects of chronic stress and potentially be used to”

Line354-356: Further develop your discussion regarding the AL scores based on assumed costs of becoming dominant.

Line 361: Revise sentence: “Thus, there is an opportunity re-assess previously measured data as a composite score”

Line 372: Example of revision of punctuation required: “To make this happen biomarker discovery and advances in wildlife endocrinology are”. However, please check the whole manuscript.

Line 384-385: “AL may be become a useful measurement of stress”. This manuscript needs proper proofreading before publication.

S3 table: PRISMA should be described in the methodology and S3 should be referenced there.

6. PLOS authors have the option to publish the peer review history of their article (what does this mean?). If published, this will include your full peer review and any attached files.

Reviewer #1: No

Reviewer #2: No

Reviewer #3: **Yes: **David M. Olson, Ph.D., D.Sc., FRCOG, FCAHS

---

## [Author Response · Author response to Decision Letter 0]

5 Aug 2022

Dear PLOS One editorial and review team,

The authors would like to extend our thanks for the detailed and thoughtful reviews of our manuscript. We feel that the comments are incredibly helpful and have led to an improved version of our manuscript. Please find responses to each of the reviews below. 

Best,

Katie Seeley

Reviewer #1

Line 58:

Please add “short-term” to acute stress (as done for chronic with “prolonged” “or long-term”. I am aware this is an old discussion but I think the time aspect is important.

Response: Added this term (line 59).

Line 64 and 65:

concerning “leukocyte values”: do the others refer here to shifts in leukocyte numbers and composition (according to the citation)? Please clarify. The authors might also add leukocyte function (e.g. Huber, N., Marasco, V., Painer, J., Vetter, S. G., Göritz, F., Kaczensky, P., & Walzer, C. (2019). Leukocyte coping capacity: an integrative parameter for wildlife welfare within conservation interventions. Frontiers in veterinary science, 6, 105.)

Response: Clarification was made, thank you for the suggestion of the additional reference, this has been added as well (lines 65-66).

Line 66:

…..“there is no one measure that”….;

only a suggestion: there is no single measure….

Response: Edited as suggested (line 68).

Line 79:

….“allowing for a more flexible adaptation to stressors”.

I me be mistaken here but in my view allostasis and the two mentioned major axes allow to adapt to/cope with stressors in general (not “more” flexible). Further, stressors may not only specifically be coming from the environment but also from e.g. disrupted internal conditions independent from the environment). Apologies for being picky here.

Response: The sentence has been edited to be more precise and includes internal stressors as well as external (lines 79-82).

Line 93:

I do not really see how the listed parameters mentioned in the next few lines reflect immune function. I suggest skipping “immune” here.

Response: Edited as suggested (line 95).

Line 133:

Please add a ® behind Pubmed. Please also check for the other databases if this is required (same for the databases in the Prisma flow diagram)

Response: Added (line 136).

Lines 274, 275:

This is an important aspect pointed out here! However, for the reader (who may not be as familiar with stress physiology and the link between the neuroendocrine and the immune system and e.g. changes in movement behaviour and how this is linked to epidemiology etc.) it may not be clear how AL and diseases ecology are linked. Please add a brief explanation here.

Response: Thank you for this input. Without getting too far into the weeds, since this is a nuanced and complex topic, I added a few lines to further clarify and expand upon this point (line 288-294). 

Lines 302, 303.

“Thank you” for stressing this point!

Response: You are welcome, we are glad you agree!

Lines 309 to 311:

I absolutely agree, that using a single biomarker to assess AL is not sufficient and leads to problematic interpretations.

However: i) in my experience it is difficult to distinguish or entangle acute/short-term and chronic/long-term stress – especially when working with wildlife or animals which can only be sampled once and ii) does the initial purpose of AL and ALI to evaluate chronic stress conditions automatically invalidate its use to assess the “cost” linked to defined (known) acute stress reactions such capture or handling?

It would be helpful for the reader, if the authors could be more specific here in terms of either giving explanations to why AL/ALI should only be used in the context of its initial aim or consider that depending on the choice of biomarkers* and their interpretation AL/ALI could also be a suitable approach to assess acute stress reactions and the associated costs.

* Gormally, B. M., & Romero, L. M. (2020). What are you actually measuring? A review of techniques that integrate the stress response on distinct time‐scales. Functional Ecology, 34(10), 2030-2044.

Response: The authors thank the reviewer for their thoughtful insight into this point and for providing the incredibly useful reference. The authors agree that applying ALI to wildlife populations is fraught and requires modifications and adjustments to try and address some of the complexities of dealing with animals. A section has been added to expand upon this discussion point further (lines 344-354). 

Lines 372 to 373:

I go in line with authors on the suggested next and needed steps of biomarker discovery and advances in wildlife endocrinology. I do, however, also suggest to mention the gap of missing reference values for many wildlife species. In this context, it would also be nice if the authors mention that in many species it is only possible to obtain the necessary samples after capture and often immobilisation/anaesthesia, biasing the measured biomarkers and making their interpretation or use for AL difficult.

Response: Excellent point, the limitations of biomarker incorporation have been included (lines 406-411). 

Lines 381 to 382:

Again, I absolutely agree – but please elaborate why! researches should focus on chronic stress (maybe you could do so in the discussion above (please see comment to lines 309 to 311).

Response: Please see response for lines 309- 311 above which address this comment as well. 

Lines 384 to 386:

Although I personally are in line with the authors, again, the statement could be better justified in the discussion above, (link AL/ALI to welfare, which is straighter forward to understand – but please elaborate on the link to disease and why ALIs could be used as a predictor).

Response: For the purpose of clarity the authors chose to remove this portion of the text. ALI is not yet used on an individual basis in human populations so trying to explain the nuances of poplution levels applications to determine at risk populations may be a bit beyond the scope of this manuscript.

Line 403:

In accordance to my previous comment on the discussion of AL/ALI in the context of acute and chronic stress. Why do the authors see this as problematic?

Response: edits have been made it the discussion to try and further illustrate that the initial purpose of ALI in humans is to evaluate the impact of chronic stressors, which is why an ALI includes an array of biomarkers, some of which increase acutely but others that would only change over a longer time scale with continued dysregulation (lines 336-339). Additional discussion about the potential utility of a matrix of acute stress biomarkers has been added as well (lines 350-354).

Reviewer #2: 

In general, the manuscript is well-written and appears thorough.

The following are minor errors:

Line 138: Change "complimentary" to "complementary" (as in "complementary angle").

Response: Edited as suggested (line 141).

Line 141: Change "to being" to "to be".

Response: Edited as suggested (line 144).

Line 157: Change "thesis" to "theses".

Response: Edited as suggested (line 162).

Line 161: Change "Any article" to "Any articles" or change "were" to "was".

Response: Edited as suggested (line 173).

Line 293: The sentence "Several papers... trout [51]." appears somewhat income

Response: Edited sentence so it makes more sense (lines 312-314).

Line 299: Change from "data needs" to "data need".

Response: Edited as suggested (line 318).

Line 350: Change "Authors" to "The authors".

Response: Edited as suggested (line 365).

Reviewer #3: 

Keywords: allostasis; Allostatic load; allostatic load index; animals; Stress - change capital words

Response: Edits were made as suggested in PLOS One

Line 96: waist–hip ratio, and glycosylated hemoglobin (HbA1c). Please add comma after waist-hip ratio to be consistent with your comma style.

Response: Edited as suggested (line 98).

LINE 118: The authors should briefly cite the tools used to conduct the scoping review (e.g. PRISMA, Arksey and O’Malley’s original framework)

Response: Thank you for this comment, this portion of the methodology has been better described (lines 120-121).

Line 132: Have you used any reference management software? Please include.- 

Response: We did use reference management software (Zotero) and that information has been added to the manuscript (line 168). 

Line 143-147: Please include semicolons to divide the journals included in the hand-search. The semicolon will help readers to keep track of the journal`s names.

Response: Edited as suggested (line 146-150).

Line 153: “An initial screening of the title and abstract was conducted”. Please add the number of authors that participated in each screening step and their initials.

Response: Initials have been added (lines 124-125, 161, 168, and 179)

Line 172: correct heartrate to heart rate throughout the paper.

Response: Correction made throughout the manuscript.

Line 218: Fig 2. Please add * before “Search results for 2021 were through June and so only represent half of the publications from this year.”

Response: Edited as suggested (line 226).

Line 221: Fig 3. Please include the numbers found for each taxa in the figure legend.

Response: The graphic was edited to include the numbers of each column on the graph itself for increased clarity to the reader (See Fig 3). 

Table 1: revise sentence: “No hemolymph biomarkers were but muscle dehydration was

associated with AL in salininity of 30%.” – Freshwater prawn (Macrobrachium acanthurus).

Response: Edited as suggested in table 1.

“Fish exposed to highest aluminum concentrations didn’t mount an appropriate cortisol ..” - Atlantic Salmon (Salmo salar L.). Contraction words should not be used.

Response: Edited as suggested in table 1.

Line 294: Please revise spaces between the text and reference as : “hyena[99], Assamese macaques [74], bearded capuchins [80], cichlids [59] and rainbow trout [51]..” Also, please check when the first letter should be capitalized or not as in Assamese macaques.

Response: We have revised the spaces here and throughout the manuscript. We have kept “Assamese” capitalized since Assamese is a proper noun.

Revise scientific English and punctuation throughout the paper. (e.g. “Similarly, Hing et al investigated the effects that wildfires have on brush-tailed bettongs (Bettongia penicillata), by measuring fecal glucocorticoids two days after a fire.”

Response: Thank you for you input regarding grammar and punctuation. The manuscript has been closely reviewed by all three authors as well as an outside reviewer prior to resubmission.

Line 324: Be consistent in the abbreviations: AL - allostatic load “put conservation mitigation actions into place, neither study adequately assessed allostatic load”

Response: The manuscript was reviewed to ensure consistency in abbreviations throughout the text.

The reader could wonder what the actions/roles of each AL mediator are. I suggest creating a table describing the category each mediator fits. For instance: cytokines: immune system, pro- or anti-inflammatory actions, etc.

Response: This is an excellent suggestion. However, the authors feel that this is beyond the scope of this review due to the number of biomarkers that have been included in allostatic load indices. Instead, references for publications describing the function of the various biomarkers are provided in the manuscript for readers who are looking for more details. These references can be found in lines 70 and 102). 

Line 342: Please add the Greek symbol in all interleukin-1b (IL-1b) references in the paper.

Response: Edits made throughout the manuscript.

Line 346: Missing word. Revise sentence: “concluded that the rCALM index estimated the effects of chronic stress and potentially be used to”

Response: Edited as suggested (line 226).

Line354-356: Further develop your discussion regarding the AL scores based on assumed costs of becoming dominant.

Response: We expanded on this discussion point to further highlight the limitations in the way that AL was assessed in this publication (line 380-387).

Line 361: Revise sentence: “Thus, there is an opportunity re-assess previously measured data as a composite score”

Response: Edited as suggested (line 390).

Line 372: Example of revision of punctuation required: “To make this happen biomarker discovery and advances in wildlife endocrinology are”. However, please check the whole manuscript.

Response: The punctuation in the manuscript has been revised here and elsewhere. 

Line 384-385: “AL may be become a useful measurement of stress”. This manuscript needs proper proofreading before publication.

Response: Thank you for you input regarding grammar and punctuation. 

S3 table: PRISMA should be described in the methodology and S3 should be referenced there.

Response: This has been added to the methodology section and supplemental numbering has been re-ordered accordingly (lines 120-125).

---

## [Decision Letter · Decision Letter 1]

17 Aug 2022

The application of allostasis and allostatic load in animal species: A scoping review

PONE-D-22-08746R1

Dear Dr. Seeley,

We’re pleased to inform you that your manuscript has been judged scientifically suitable for publication and will be formally accepted for publication once it meets all outstanding technical requirements.

Kind regards,

Sylvain Giroud, PhD

Academic Editor

PLOS ONE

Reviewers' comments:

Reviewer's Responses to Questions

**Comments to the Author**

1. If the authors have adequately addressed your comments raised in a previous round of review and you feel that this manuscript is now acceptable for publication, you may indicate that here to bypass the “Comments to the Author” section, enter your conflict of interest statement in the “Confidential to Editor” section, and submit your "Accept" recommendation.

Reviewer #1: All comments have been addressed

2. Is the manuscript technically sound, and do the data support the conclusions?

Reviewer #1: Yes

3. Has the statistical analysis been performed appropriately and rigorously? 

Reviewer #1: Yes

4. Have the authors made all data underlying the findings in their manuscript fully available?

Reviewer #1: Yes

5. Is the manuscript presented in an intelligible fashion and written in standard English?

Reviewer #1: Yes

6. Review Comments to the Author

Reviewer #1: (No Response)

7. PLOS authors have the option to publish the peer review history of their article (what does this mean?). If published, this will include your full peer review and any attached files.

Reviewer #1: **Yes: **Nikolaus Huber, DVM, PhD

---

## [Editor Report · Acceptance letter]

19 Aug 2022

PONE-D-22-08746R1 

The application of allostasis and allostatic load in animal species: A scoping review 

Dear Dr. Seeley:

I'm pleased to inform you that your manuscript has been deemed suitable for publication in PLOS ONE. Congratulations! Your manuscript is now with our production department. 

Kind regards, 

on behalf of

Dr. Sylvain Giroud 

Academic Editor

PLOS ONE